# Product distribution learning with imperfect advice

**Arnab Bhattacharyya**
Department of Computer Science
University of Warwick
arnab.bhattacharyya@warwick.ac.uk

**Davin Choo**
Harvard John A. Paulson School Of Engineering And Applied Sciences
Harvard University
davinchoo@seas.harvard.edu

**Philips George John**
CNRS@CREATE & Dept. of Computer Science
National University of Singapore
philips.george.john@u.nus.edu

**Themis Gouleakis**
College of Computing & Data Science
Nanyang Technological University
themis.gouleakis@ntu.edu.sg

## Abstract

Given i.i.d. samples from an unknown distribution $P$, the goal of distribution learning is to recover the parameters of a distribution that is close to $P$. When $P$ belongs to the class of product distributions on the Boolean hypercube $\{0, 1\}^d$, it is known that $\Omega(d/\varepsilon^2)$ samples are necessary to learn $P$ within total variation (TV) distance $\epsilon$. We revisit this problem when the learner is also given as advice the parameters of a product distribution $Q$. We show that there is an efficient algorithm to learn $P$ within TV distance $\varepsilon$ that has sample complexity $\tilde{O}(d^{1-\eta}/\varepsilon^2)$, if $\|\mathbf{p} - \mathbf{q}\|_1 < \varepsilon d^{0.5-\Omega(\eta)}$. Here, $\mathbf{p}$ and $\mathbf{q}$ are the mean vectors of $P$ and $Q$ respectively, and no bound on $\|\mathbf{p} - \mathbf{q}\|_1$ is known to the algorithm a priori.

## 1 Introduction

Science fundamentally relies on the ability to learn models from data. In many real-world settings, the majority of available datasets consist of unlabeled examples – sample points drawn without corresponding labels, outputs or classifications. These unlabeled datasets are often modeled as samples from a joint probability distribution on a large domain. The goal of *distribution learning* is to output the description of a distribution that approximates the underlying distribution that generated the observed samples. See [Dia16] for a comprehensive survey.

In practice, distribution learning rarely occurs in isolation. While a given dataset may be new, one often has access to previously learned models from related datasets. Alternatively, the data may arise from an evolving process, motivating the reuse of information from past learning to guide current inference. This prior information can be viewed as a form of "advice" or "prediction" of some kind. In the framework of algorithms with predictions, the objective is to integrate such advice in a way that improves performance when the advice is accurate, while ensuring robustness: performance should not degrade beyond that an advice-free baseline algorithm, even when the predictions are inaccurate. Most previous works in this setting are in the context of online algorithms, e.g. for the ski-rental problem [GP19, WLW20, ADJ+20], non-clairvoyant scheduling [PSK18], scheduling [LLMV20, BMRS20, AJS22], augmenting classical data structures with predictions (e.g. indexing [KBC+18] and Bloom filters [Mit18]), online selection and matching problems [AGKK20,

DLPLV21, CGLB24, CJS25], online TSP [BLMS$^+$22, GLS23], and a more general framework of online primal-dual algorithms [BMS20]. However, there have been some recent applications to other areas, e.g. graph algorithms [CSVZ22, DIL$^+$21], causal learning [CGB23], mechanism design [GKST22, ABG$^+$22], and most relevantly to us, distribution learning [BCGG25].

In this work, we study the problem of learning *product distributions* over the $d$-dimensional Boolean hypercube $\{0,1\}^d$, arguably one of the most fundamental classes of discrete high-dimensional distributions. A product distribution $P$ is fully specified by its *mean vector* $\mathbf{p} \in [0,1]^d$, where the $i$-th coordinate $\mathbf{p}_i$ represents the expectation of the $i$-th marginal of $P$, or equivalently, the probability that the $i$-th coordinate of a sample from $P$ is 1. It is well-known that $\Theta(d/\varepsilon^2)$ samples from a product distribution $P$ are both necessary and sufficient to learn a distribution $\widehat{P}$ such that $\mathrm{d}_{\mathrm{TV}}(P,\widehat{P}) \le \varepsilon$ with probability at least $2/3$, where $\mathrm{d}_{\mathrm{TV}}$ denotes the total variation distance. This optimal sample complexity is achieved by a simple, natural and efficient algorithm: computing the empirical mean of each coordinate. Motivated by the framework of algorithms with predictions, we investigate whether this sample complexity can be improved when, in addition to samples from $P$, the learner is given an advice mean vector $\mathbf{q} \in [0,1]^d$. Importantly, we make no assumption that $\mathbf{q}$ is close to the true mean $\mathbf{p}$. However, if we can detect that $\mathbf{q}$ is accurate – i.e., that $\|\mathbf{q}-\mathbf{p}\|$ is small in an appropriate norm – can this information be leveraged to constrain the search space and improve sample or computational efficiency? Our goal is to design algorithms that adapt to the quality of the advice: performing better when $\mathbf{q}$ is accurate, while remaining robust when it is not.

Our main result establishes that this is indeed possible. Specifically, we show that if $\|\mathbf{q}-\mathbf{p}\|_1 \ll \varepsilon\sqrt{d}$, then there exists a polynomial-time algorithm with sample complexity *sublinear in $d$* that outputs a distribution $\widehat{P}$ such that $\mathrm{d}_{\mathrm{TV}}(P,\widehat{P}) \le \varepsilon$ with probability at least $2/3$. More precisely, under the regularity condition that no coordinate of $P$ is too close to deterministic (i.e., bounded away from 0 and 1), we show that the sample complexity is:

$$\tilde{O}\left( \frac{d}{\varepsilon^2}\left( d^{-\eta} + \min\left(1, \frac{\|\mathbf{p}-\mathbf{q}\|_1^2}{d^{1-4\eta}\varepsilon^2}\right)\right)\right)$$

for any small enough constant $\eta$. In particular, when $\|\mathbf{p}-\mathbf{q}\|_1$ is small, the dependence on $d$ becomes sublinear. We also prove that the non-determinism assumption is necessary: if coordinates of $P$ can be arbitrarily close to 0 or 1, then sample complexity that is sublinear in $d$ is impossible, even when $\|\mathbf{q}-\mathbf{p}\|_1 = O(1)$. Furthermore, we show that when $\|\mathbf{q}-\mathbf{p}\|_1 \gg \varepsilon\sqrt{d}$, no algorithm with sublinear sample complexity exists.

## 1.1 Technical Overview

We call a product distribution $P$ *balanced* if no marginal of $P$ is too biased. That is, each coordinate of $\mathbf{p}$ is bounded away from 0 and 1. It is known that, for balanced distributions, learning $P$ in TV distance is equivalent to learning the mean vector $\mathbf{p}$ with $\ell_2$-error. Hence, in this overview, we focus on the latter task.

To build intuition about how an advice vector $\mathbf{q}$ can be exploited, consider the following two situations:

1. **Exact advice**: Suppose $\mathbf{q} = \mathbf{p}$. Then, it suffices to verify that $\|\mathbf{q}-\mathbf{p}\|_2 \le \varepsilon$ and a learning algorithm can simply return $\mathbf{q}$. This is the classic *identity testing* problem, which has been extensively studied (see [Can20] for a detailed survey). For product distributions, Daskalakis and Pan [DP17] and Canonne, Diakonikolas, Kane and Stewart [CDKS17] independently showed that identity testing requires $\Theta(\sqrt{d}/\varepsilon^2)$ samples. This demonstrates that sublinear sample complexity is achievable when $\mathbf{q} = \mathbf{p}$. Morever, $\Omega(\sqrt{d}/\varepsilon^2)$ is a fundamental lower bound that applies even when $\mathbf{q} = \mathbf{p}$.

2. **Sparse disagreement**: Suppose $\mathbf{q}$ differs from $\mathbf{p}$ in at most $t$ coordinates, i.e. $\|\mathbf{q}-\mathbf{p}\|_0 \le t$. In this case, one only needs to estimate $\mathbf{p}$ on those $t$ coordinates, so the information-theoretic sample complexity should scale as $\sim \log\binom{d}{t}$. Unfortunately, since $t$ is unknown a priori, we cannot directly exploit this sparsity. However, from the compressive sensing literature, it is known that closeness in $\ell_2$ norm to a $t$-sparse vector can be certified by a small $\ell_1$ norm (e.g., Theorem 2.5 of [FR13]).

Motivated by the above, we take the quality of the advice $\mathbf{q}$ to be governed by the $\ell_1$-distance $\|\mathbf{p}-\mathbf{q}\|_1$, and aim for an algorithm whose sample complexity improves as this distance decreases.

Suppose we can certify that $\|\mathbf{p} - \mathbf{q}\|_1 \leq \lambda$. Then, we may restrict attention to the $\ell_1$-ball of radius $\lambda$ centered at $\mathbf{q}$, and cover it with $N$ $\ell_2$-balls of radius $\varepsilon$, where the covering number $N$ grows polynomially as $d^{O(\lambda^2/\varepsilon^2)}$. It is known (e.g. see Chapter 4 of [DL01]) that using the Scheffé tournament method, the sample complexity of learning $\mathbf{p}$ up to $\ell_2$-norm $\varepsilon$ scales as $(\log N)/\varepsilon^2$, which yields a bound of $O(\frac{\lambda^2}{\varepsilon^4} \log d)$. While the Scheffé tournament is computationally inefficient, the same sample complexity guarantee can be achieved efficiently by solving a constrained least squares problem. More precisely, given samples $\mathbf{x}_1, \ldots, \mathbf{x}_n$ from $P$, we consider the estimator:

$$\underset{\mathbf{b} \in \mathbb{R}^d : \|\mathbf{b} - \mathbf{q}\|_1 \leq \lambda}{\operatorname{argmin}} \frac{1}{n} \sum_{i=1}^{n} \|\mathbf{x}_i - \mathbf{b}\|_2^2$$

For $n = O(\lambda^2 \varepsilon^{-4} \log d)$, this estimates achieves $\ell_2$-error at most $\varepsilon$.

The key challenge that remains is then to approximate $\lambda \approx \|\mathbf{p} - \mathbf{q}\|_1$ using a sublinear number of samples from $P$. To this end, we devise a new identity testing algorithm that, using $O(\sqrt{d}/\varepsilon^2)$ samples, either (i) 2-approximates $\|\mathbf{p} - \mathbf{q}\|_2$, or (ii) certifies that $\|\mathbf{p} - \mathbf{q}\|_2 \leq \varepsilon$, in which we simply return $\mathbf{q}$. If we are in case (i), we can upper bound $\|\mathbf{p} - \mathbf{q}\|_1$ with $\lambda = \|\mathbf{p} - \mathbf{q}\|_2 \cdot \sqrt{d}$. However, this would make the sample complexity of the learning algorithm to be $O\left(\frac{\lambda^2}{\varepsilon^4} \log d\right) = O\left(\frac{d \log(d) \cdot \|\mathbf{p} - \mathbf{q}\|_2^2}{\varepsilon^4}\right) \gg \frac{d}{\varepsilon^2}$, i.e., exceeding the standard $O(d/\varepsilon^2)$ bound and defeating the purpose. To improve upon this, we can partition the $d$ coordinates into $d/k$ blocks of size $k$ each. Then, within each block, the ratio between the $\ell_1$ and $\ell_2$ norms improves from $\sqrt{d}$ to $\sqrt{k}$. By appropriately choosing $k$, we can obtain a non-trivial reduction in overall sample complexity.

The structure of our algorithm described above and its analysis parallels the recent work by [BCGG25], which addresses the problem of learning Gaussian distributions with imperfect advice. However, our setting differs in several important ways:

- [BCGG25] used a well-known algorithm to approximate the $\ell_2$ norm of a Gaussian's mean vector. In contrast, our $\ell_2$-approximation algorithm for product distribution is new, to the best of our knowledge.

- We critically rely on the balancedness assumption to relate total variation distance and $\ell_2$ error of the mean vector. No such assumption is needed in the Gaussian setting. In fact, we show that for product distributions, balancedness is essential: without it, no sublinear-sample algorithm exists, even when the advice vector is $O(1)$-close to the truth in $\ell_1$ distance. We find this somewhat surprising since $O(\sqrt{d}/\varepsilon^2)$ samples suffice without any balancedness assumptions in identity testing [DP17, CDKS17].

## 2 Preliminaries

A distribution $P$ on $\{0,1\}^d$ is said to be a *product distribution* if there exist distributions $P_1, \ldots, P_d$ on $\{0,1\}$ such that $P(\mathbf{x}) = P_1(x_1) \cdot P_2(x_2) \cdots P_d(x_d)$ for every $x \in \{0,1\}^d$. In this case, we can write $P = P_1 \otimes P_2 \cdots \otimes P_d$.

**Definition 2.1** (Mean vectors). The mean vector of a distribution $P$ is $\mathbf{p} \triangleq \mathbb{E}_{x \sim P}[x]$. In particular, if $P = P_1 \otimes \cdots \otimes P_d$ is a product distribution, $\mathbf{p} = [p_1 \quad \cdots \quad p_d]$, where $p_i = P_i(1)$.

For a vector $\mathbf{p} \in [0,1]^d$, we denote by $\mathrm{Ber}(\mathbf{p})$ the product distribution with mean vector $\mathbf{p}$. We next define the notion of balancedness.

**Definition 2.2.** For $\tau \in [0, 1/2]$, a product distribution $P$ on $\{0,1\}^d$ is said to be $\tau$-*balanced* if for every $i \in [d]$, the marginal $P_i$ satisfies $\tau \leq P_i(1) \leq 1 - \tau$.

**Proposition 2.3** (e.g., [CDKS17], Lemma 1). *Suppose $P$ and $Q$ are $\tau$-balanced product distributions on $\{0,1\}^d$ with mean vectors $\mathbf{p}$ and $\mathbf{q}$ respectively. Then their KL divergence $\mathrm{d}_{\mathrm{KL}}(P\|Q)$ satisfies:*

$$2\|\mathbf{p} - \mathbf{q}\|_2^2 \leq \mathrm{d}_{\mathrm{KL}}(P\|Q) \leq \frac{2}{\tau}\|\mathbf{p} - \mathbf{q}\|_2^2.$$

**Proposition 2.4.** *Suppose $P$ and $Q$ are $\tau$-balanced product distributions on $\{0,1\}^d$ with mean vectors $\mathbf{p}$ and $\mathbf{q}$ respectively. Then, for some constant $c < 0.2$, the TV distance $\mathrm{d}_{\mathrm{TV}}(P,Q)$ satisfies:*

$$c \cdot \min\{1, \|\mathbf{p} - \mathbf{q}\|_2\} \leq \mathrm{d}_{\mathrm{TV}}(P,Q) \leq \frac{1}{\sqrt{\tau}}\|\mathbf{p} - \mathbf{q}\|_2.$$

*Proof.* The first inquality is the main result of [Kon25]. The second inequality follows from applying Pinsker to the upper bound on $d_{KL}$ in Proposition 2.3. □

Note that the dependence on $\tau$ above is necessary (up to constant factors). For example, suppose $P = \text{Ber}(\mathbf{0})$ and $Q = \text{Ber}(\mathbf{u})$ where $\mathbf{0}$ is the all-zero vector and $\mathbf{u} = [\frac{1}{d}, \frac{1}{d}, \dots, \frac{1}{d}]$. Then, $\|\mathbf{0} - \mathbf{u}\|_2 = 1/\sqrt{d}$, while $d_{TV}(P, Q) \geq P(\mathbf{0}) - Q(\mathbf{0}) = 1 - (1 - 1/d)^d \approx 1 - 1/e$.

## 3   Algorithm

The goal of this section is to establish the following result.

**Theorem 3.1.** *There exists algorithm* TESTANDOPTIMIZEMEAN *that for any given* $\varepsilon, \delta, \tau \in (0, 1)$, $\eta \geq 0$, *and* $\mathbf{q} \in [0, 1]^d$, *and sample access to a* $\tau$-*balanced product distribution* $\text{Ber}(\mathbf{p})$ *on* $\{0, 1\}^d$, *it draws* $n = \tilde{O}\left(\frac{d}{\varepsilon^2} \cdot (d^{-\eta} + \min\{1, f(\mathbf{p}, \mathbf{q}, d, \eta, \varepsilon)\})\right)$ *i.i.d. samples from* $\text{Ber}(\mathbf{p})$, *where:*
$$f(\mathbf{p}, \mathbf{q}, d, \eta, \varepsilon) = \frac{\|\mathbf{p} - \mathbf{q}\|_1^2}{d^{1-4\eta}\tau^6\varepsilon^2}.$$
*The algorithm produces as output* $\widehat{\mathbf{p}}$ *in* $\text{poly}(n, d)$ *time such that* $d_{TV}(\text{Ber}(\mathbf{p}), \text{Ber}(\widehat{\mathbf{p}})) \leq \varepsilon$ *with success probability at least* $1 - \delta$.

A basic component of the algorithm is a test to determine how close the advice $\mathbf{q}$ is to the true $\mathbf{p}$ in $\ell_2$ norm.

**Lemma 3.2** (Tolerant mean tester). *Given* $\varepsilon > 0$, $\delta \in (0, 1)$, $d$ *sufficiently large integer, and* $\mathbf{q} \in [0, 1]^d$, *there is a tolerant tester* TMT *that uses* $O\left(\frac{\sqrt{d}}{\varepsilon^2} \log\left(\frac{1}{\delta}\right)\right)$ *i.i.d. samples from* $\text{Ber}(\mathbf{p})$ *and satisfies both conditions below with probability at least* $1 - \delta$:

1. *If* $\|\mathbf{p} - \mathbf{q}\|_2 \leq \varepsilon$, *then the tester outputs* Accept

2. *If* $\|\mathbf{p} - \mathbf{q}\|_2 \geq 2\varepsilon$, *then the tester outputs* Reject

*Proof.* Notice that if $\sum p_i = \sum q_i = 1$, we could interpret $p_1, \dots, p_d$ and $q_1, \dots, q_d$ as distributions $\tilde{\mathbf{p}}$ and $\tilde{\mathbf{q}}$ on $[d]$ that sample $i \in [d]$ with probability $p_i$ and $q_i$ respectively. Diakonikolas and Kane [DK16] showed that using $O(\|\tilde{\mathbf{p}}\|_2/\varepsilon^2)$ samples from $\tilde{\mathbf{p}}$, one can test whether $\|\tilde{\mathbf{p}} - \tilde{\mathbf{q}}\|_2 \leq \varepsilon$ or $\|\tilde{\mathbf{p}} - \tilde{\mathbf{q}}\|_2 \geq 2\varepsilon$. Inspired by this observation, we mimic the analysis of [DK16] to devise a tolerant tester for general product distributions.

Assume the desired failure probability to be $1/3$; we can reduce to any $\delta$ by repeating the test $O(\log 1/\delta)$ times and taking the majority vote. Set $m = c\sqrt{d}/\varepsilon^2$ for a large enough constant $c$, and let $m_i$ be sampled independently from $\text{Poi}(m)$ for each $i \in [d]$. Note that $\max_i m_i \leq 2em$ with high probability; we condition on this event and set the desired failure probability to $1/4$. Therefore, using $2em$ samples from $\text{Ber}(\mathbf{p})$, for each $i$, we can obtain $m_i$ samples from the $i$th coordinate, and let $X_i$ be the number of times the $i$'th coordinate is sampled to be 1. Note that by standard properties of the Poisson distribution, the $X_1, \dots, X_d$ are independent and each $X_i$ is sampled from $\text{Poi}(mp_i)$.

Define the statistic $Z = \sum_{i=1}^d Z_i$, where:
$$Z_i = (X_i - mq_i)^2 - X_i.$$
Using similar calculations as in [DK16], we can show that:
$$\mathbb{E}[Z_i] = m^2(p_i - q_i)^2 \qquad \text{and} \qquad \mathbb{E}[Z] = m^2\|\mathbf{p} - \mathbf{q}\|_2^2.$$
Also, we can calculate the variance to be:
$$\text{Var}[Z] = 4m^3 \sum_{i=1}^d p_i(p_i - q_i)^2 + 2m^2 \sum_{i=1}^d p_i^2 \leq 4m^3\|\mathbf{p}\|_2\|\mathbf{p} - \mathbf{q}\|_4^2 + 2m^2\|\mathbf{p}\|_2^2,$$
where the inequality is by Cauchy-Schwarz.

If $\|\mathbf{p} - \mathbf{q}\|_2 \leq \varepsilon$, then $\mathbb{E}[Z] \leq c^2 d/\varepsilon^2$, and $\text{Var}[Z] \leq (4c^3 + 2c^2)d^2/\varepsilon^4$. On the other hand, it always holds that $\mathbb{E}[Z] \geq c^2 d\|\mathbf{p} - \mathbf{q}\|_2^2/\varepsilon^4$, and $\text{Var}[Z] \leq (4c^3 d^{1.5}\|\mathbf{p}\|_2\|\mathbf{p} - \mathbf{q}\|_2^2/\varepsilon^2 + 2c^2 d\|\mathbf{p}\|_2^2)/\varepsilon^4 \leq (4c^3\|\mathbf{p} - \mathbf{q}\|_2^2/\varepsilon^2 + 2c^2)d^2/\varepsilon^4$, since $\|\mathbf{p}\|_2 \leq \sqrt{d}$. Using Chebyshev's inequality, if $c$ is large enough, when $\|\mathbf{p} - \mathbf{q}\|_2 \leq \varepsilon$, $Z$ is at most $2c^2 d/\varepsilon^2$ with probability $3/4$, but when $\|\mathbf{p} - \mathbf{q}\|_2 \geq 2\varepsilon$, $Z$ is at least $3c^2 d/\varepsilon^2$ with probability $3/4$. □

**Algorithm 1** The APPROXL1 algorithm.

---

1: **Input**: Block size $k \in [d]$, lower bound $\alpha > 0$, upper bound $\zeta > 2\alpha$, failure rate $\delta \in (0, 1)$, advice $\mathbf{q} \in [0, 1]^d$, and i.i.d. samples $\mathcal{S}$ (multiset) from $\mathrm{Ber}(\mathbf{p})$.
2: **Output**: Fail or $\lambda \in \mathbb{R}$
3: Define $w = \lceil d/k \rceil$ and $\delta' = \frac{\delta}{w \cdot \lceil \log_2 \zeta/\alpha \rceil}$
4: Partition the index set $[d]$ into $w$ blocks:

$$\mathbf{B}_1 = \{1, \ldots, k\}, \mathbf{B}_2 = \{k + 1, \ldots, 2k\}, \ldots, \mathbf{B}_w = \{k(w - 1) + 1, \ldots, d\}$$

5: **for** $j \in \{1, \ldots, w\}$ **do**
6:      Define multiset $\mathcal{S}_j = \{\mathbf{x}_{\mathbf{B}_j} \in \mathbb{R}^{|\mathbf{B}_j|} : \mathbf{x} \in \mathcal{S}\}$ as the samples projected to $\mathbf{B}_j$
7:      Let $\mathbf{q}_{\mathbf{B}_j} \in \mathbb{R}^{|\mathbf{B}_j|}$ be the vector $\mathbf{q}$ projected to coordinates $\mathbf{B}_j$.
8:      Initialize $o_j = \mathsf{Fail}$
9:      **for** $i = 1, 2, \ldots, \lceil \log_2 \zeta/\alpha \rceil$ **do**
10:          Define $l_i = 2^{i-1} \cdot \alpha$
11:          Let $\mathtt{Outcome}$ be the output of the tolerant tester TMT of Lemma 3.2 using sample set $\mathcal{S}_j$
12:          with parameters $\varepsilon \leftarrow l_i, \delta \leftarrow \delta', d \leftarrow |\mathbf{B}_j|$ and $\mathbf{q} \leftarrow \mathbf{q}_{\mathbf{B}_j}$
13:          **if** $\mathtt{Outcome}$ is Accept **then**
14:              Set $o_j = l_i$ and **break** {Escape inner loop for block $j$}
15:          **end if**
16:      **end for**
17: **end for**
18: **if** there exists a Fail amongst $\{o_1, \ldots, o_w\}$ **then**
19:      **return** Fail
20: **else**
21:      **return** $\lambda = 2 \sum_{j=1}^{w} \sqrt{|\mathbf{B}_j|} \cdot o_j$ {$\lambda$ is an estimate for $\|\mathbf{p} - \mathbf{q}\|_1$}
22: **end if**

---

**Lemma 3.3.** *Let $k$, $\alpha$, and $\zeta$ be the input parameters to the* APPROXL1 *algorithm. Given $q \in [0, 1]^d$ and $m(k, \alpha, \delta) \triangleq \lceil \frac{16\sqrt{k}}{3\alpha^2} \rceil \cdot \left(1 + \lceil \log\left(\frac{12 w \cdot \log_2 \lceil \zeta/\alpha \rceil}{\delta}\right) \rceil\right)$ i.i.d. samples from $\mathrm{Ber}(\mathbf{p})$,* APPROXL1 *succeeds with probability at least $1 - \delta$ and has the following properties:*
*Property 1: If* APPROXL1 *outputs* Fail*, then $\|\mathbf{p} - \mathbf{q}\|_2 > \zeta/2$.*
*Property 2: If* APPROXL1 *outputs $\lambda \in \mathbb{R}$, then $\|\mathbf{p} - \mathbf{q}\|_1 \leq \lambda \leq 2\sqrt{k} \cdot (\lceil d/k \rceil \cdot \alpha + 2\|\mathbf{p} - \mathbf{q}\|_1)$.*

*Proof.* We begin by stating some properties of $o_1, \ldots, o_w$. Fix an arbitrary index $j \in \{1, \ldots, w\}$ and suppose $o_j$ is *not* a Fail, i.e. the tolerant tester of Lemma 3.2 outputs Accept for some $i^* \in \{1, 2, \ldots, \lceil \log_2 \zeta/\alpha \rceil\}$. Note that APPROXL1 sets $o_j = \ell_{i^*}$ and the tester outputs Reject for all smaller indices $i \in \{1, \ldots, i^* - 1\}$. Since the tester outputs Accept for $i^*$, we have that $\|\mathbf{p}_{\mathbf{B}_j} - \mathbf{q}_{\mathbf{B}_j}\|_2 \leq 2\ell_{i^*} = 2o_j$. Meanwhile, if $i^* > 1$, then $\|\mathbf{p}_{\mathbf{B}_j} - \mathbf{q}_{\mathbf{B}_j}\|_2 > \ell_{i^*-1} = \ell_{i^*}/2 = o_j/2$ since the tester outputs Reject for $i^* - 1$. Thus, we see that

- When $o_j$ is not Fail, we have $\|\mathbf{p}_{\mathbf{B}_j} - \mathbf{q}_{\mathbf{B}_j}\|_2 \leq 2o_j$.

- When $\|\mathbf{p}_{\mathbf{B}_j} - \mathbf{q}_{\mathbf{B}_j}\|_2 \leq 2\alpha$, we have $i^* = 1$ and $o_j = \ell_1 = \alpha$.

- When $\|\mathbf{p}_{\mathbf{B}_j} - \mathbf{q}_{\mathbf{B}_j}\|_2 > 2\alpha = 2\ell_1$, we have $i^* > 1$ and so $o_j < 2\|\mathbf{p}_{\mathbf{B}_j} - \mathbf{q}_{\mathbf{B}_j}\|_2$.

**Success probability.** Fix an arbitrary index $i \in \{1, 2, \ldots, \lceil \log_2 \zeta/\alpha \rceil\}$ with $\ell_i = 2^{i-1}\alpha$, where $\ell_i \leq \ell_1 = \alpha$ for any $i$. We invoke the tolerant tester with $\varepsilon = \ell_i$ in the $i^{\text{th}}$ invocation, so the $i^{\text{th}}$ invocation uses at most $m_1(k, \varepsilon, \delta) \triangleq n_{k,\varepsilon} \cdot r_\delta$ i.i.d. samples to succeed with probability at least $1 - \delta$, where $n_{k,\varepsilon} \triangleq \lceil \frac{16\sqrt{k}}{3\varepsilon^2} \rceil$ and $r_\delta \triangleq 1 + \lceil \log(12/\delta) \rceil$.

So, with at most $m(k, \alpha, \delta) \triangleq m_1(k, \alpha, \delta') = n_{k,\alpha} \cdot r_{\delta'}$ samples, *any* call to the tolerant tester succeeds with probability at least $1 - \delta'$, where $\delta' \triangleq \frac{\delta}{w \cdot \lceil \log_2 \zeta/\alpha \rceil}$. By construction, there will be at most $w \cdot \lceil \log_2 \zeta/\alpha \rceil$ calls to the tolerant tester. Therefore, by union bound, *all* calls to the tolerant tester *jointly succeed* with probability at least $1 - \delta$.

**Proof of Property 1.** When APPROXL1 outputs Fail, there exists a Fail amongst $\{o_1, \ldots, o_w\}$. For any fixed index $j \in \{1, \ldots, w\}$, this can only happen when all calls to the tolerant tester outputs Reject. This means that $\|x_{\mathbf{B}_j}\|_2 > \varepsilon_1 = \ell_i = 2^{i-1} \cdot \alpha$ for all $i \in \{1, 2, \ldots, \lceil \log_2 \zeta/\alpha \rceil\}$. In particular, this means that $\|x_{\mathbf{B}_j}\|_2 > \zeta/2$.

**Proof of Property 2.** When APPROXL1 outputs $\lambda = 2\sum_{j=1}^{w} \sqrt{|\mathbf{B}_j|} \cdot o_j \in \mathbb{R}$, we can lower bound $\lambda$ as follows:

$$\lambda = 2\sum_{j=1}^{w} \sqrt{|\mathbf{B}_j|} \cdot o_j \geq 2\sum_{j=1}^{w} \sqrt{|\mathbf{B}_j|} \cdot \frac{\|\mathbf{p}_{\mathbf{B}_j} - \mathbf{q}_{\mathbf{B}_j}\|_2}{2} \qquad \text{(since } \|\mathbf{p}_{\mathbf{B}_j} - \mathbf{q}_{\mathbf{B}_j}\|_2 \leq 2o_j)$$

$$\geq \sum_{j=1}^{w} \|\mathbf{p}_{\mathbf{B}_j} - \mathbf{q}_{\mathbf{B}_j}\|_1 \quad \text{(since } \|\mathbf{p}_{\mathbf{B}_j} - \mathbf{q}_{\mathbf{B}_j}\|_1 \leq \sqrt{|\mathbf{B}_j|} \cdot \|\mathbf{p}_{\mathbf{B}_j} - \mathbf{q}_{\mathbf{B}_j}\|_2)$$

$$= \|\mathbf{p} - \mathbf{q}\|_1 \qquad \text{(since } \sum_{j=1}^{w} \|\mathbf{p}_{\mathbf{B}_j} - \mathbf{q}_{\mathbf{B}_j}\|_1 = \|\mathbf{p}_{\mathbf{B}_j} - \mathbf{q}_{\mathbf{B}_j}\|_1)$$

That is, $\lambda \geq \|\mathbf{p} - \mathbf{q}\|_1$. Meanwhile, we can also upper bound $\lambda$ as follows:

$$\lambda = 2\sum_{j=1}^{w} \sqrt{|\mathbf{B}_j|} \cdot o_j \leq 2\sqrt{k} \sum_{j=1}^{w} o_j \qquad \text{(since } |\mathbf{B}_j| \leq k)$$

$$= 2\sqrt{k} \cdot \left( \sum_{\substack{j=1 \\ \|\mathbf{p}_{\mathbf{B}_j}-\mathbf{q}_{\mathbf{B}_j}\|_2 \leq 2\alpha}}^{w} o_j + \sum_{\substack{j=1 \\ \|\mathbf{p}_{\mathbf{B}_j}-\mathbf{q}_{\mathbf{B}_j}\|_2 > 2\alpha}}^{w} o_j \right)$$

$$\text{(partitioning the blocks based on } \|\mathbf{p}_{\mathbf{B}_j} - \mathbf{q}_{\mathbf{B}_j}\|_2 \text{ versus } 2\alpha)$$

$$= 2\sqrt{k} \cdot \left( \sum_{\substack{j=1 \\ \|\mathbf{p}_{\mathbf{B}_j}-\mathbf{q}_{\mathbf{B}_j}\|_2 \leq 2\alpha}}^{w} \alpha + \sum_{\substack{j=1 \\ \|\mathbf{p}_{\mathbf{B}_j}-\mathbf{q}_{\mathbf{B}_j}\|_2 > 2\alpha}}^{w} o_j \right)$$

$$\text{(since } \|\mathbf{p}_{\mathbf{B}_j} - \mathbf{q}_{\mathbf{B}_j}\|_2 \leq 2\alpha \text{ implies } o_j = \alpha)$$

$$\leq 2\sqrt{k} \cdot \left( \sum_{\substack{j=1 \\ \|\mathbf{p}_{\mathbf{B}_j}-\mathbf{q}_{\mathbf{B}_j}\|_2 \leq 2\alpha}}^{w} \alpha + \sum_{\substack{j=1 \\ \|\mathbf{p}_{\mathbf{B}_j}-\mathbf{q}_{\mathbf{B}_j}\|_2 > 2\alpha}}^{w} 2\|\mathbf{p}_{\mathbf{B}_j} - \mathbf{q}_{\mathbf{B}_j}\|_2 \right)$$

$$\text{(since } \|\mathbf{p}_{\mathbf{B}_j} - \mathbf{q}_{\mathbf{B}_j}\|_2 > 2\alpha \text{ implies } o_j \leq 2\|\mathbf{p}_{\mathbf{B}_j} - \mathbf{q}_{\mathbf{B}_j}\|_2)$$

$$\leq 2\sqrt{k} \cdot \left( \sum_{\substack{j=1 \\ \|\mathbf{p}_{\mathbf{B}_j}-\mathbf{q}_{\mathbf{B}_j}\|_2 \leq 2\alpha}}^{w} \alpha + 2\sum_{\substack{j=1 \\ \|\mathbf{p}_{\mathbf{B}_j}-\mathbf{q}_{\mathbf{B}_j}\|_2 > 2\alpha}}^{w} \|\mathbf{p}_{\mathbf{B}_j} - \mathbf{q}_{\mathbf{B}_j}\|_1 \right)$$

$$\text{(since } \|\mathbf{p}_{\mathbf{B}_j} - \mathbf{q}_{\mathbf{B}_j}\|_2 \leq \|\mathbf{p}_{\mathbf{B}_j} - \mathbf{q}_{\mathbf{B}_j}\|_1)$$

$$\leq 2\sqrt{k} \cdot \left( \lceil d/k \rceil \cdot \alpha + 2\sum_{\substack{j=1 \\ \|\mathbf{p}_{\mathbf{B}_j}-\mathbf{q}_{\mathbf{B}_j}\|_2 > 2\alpha}}^{w} \|\mathbf{p}_{\mathbf{B}_j} - \mathbf{q}_{\mathbf{B}_j}\|_1 \right)$$

$$\text{(since } |\{j \in [w] : \mathbf{p}_{\mathbf{B}_j}\|_2 \leq 2\alpha\}| \leq w)$$

$$\leq 2\sqrt{k} \cdot (\lceil d/k \rceil \cdot \alpha + 2\|\mathbf{p} - \mathbf{q}\|_1)$$

$$\text{(since } \sum_{\substack{j=1 \\ \|\mathbf{p}_{\mathbf{B}_j}-\mathbf{q}_{\mathbf{B}_j}\|_2 > 2\alpha}}^{w} \|\mathbf{p}_{\mathbf{B}_j} - \mathbf{q}_{\mathbf{B}_j}\|_1 \leq \sum_{j=1}^{w} \|\mathbf{p}_{\mathbf{B}_j} - \mathbf{q}_{\mathbf{B}_j}\|_1 = \|\mathbf{p}_{\mathbf{B}_j} - \mathbf{q}_{\mathbf{B}_j}\|_1)$$

That is, $\lambda \leq 2\sqrt{k} \cdot (\lceil d/k \rceil \cdot \alpha + 2\|\mathbf{p} - \mathbf{q}\|_1)$. The property follows by putting together both bounds. $\qquad \square$

Now, suppose APPROXL1 tells us that $\|\mathbf{p} - \mathbf{q}\|_1 \leq r$. We can then perform a constrained LASSO to search for a candidate $\widehat{\mathbf{p}} \in [0,1]^d$ using $O(\frac{r^2}{\varepsilon^4} \log \frac{d}{\delta})$ samples from $\mathrm{Ber}(\mathbf{p})$.

**Lemma 3.4.** *Fix $d \geq 1$, $r \geq 0$, $\varepsilon, \delta > 0$, and $\mathbf{q} \in [0,1]^d$. Given $O(\frac{r^2}{\varepsilon^4} \log \frac{d}{\delta})$ samples from $\mathrm{Ber}(\mathbf{p})$ for some unknown $\mathbf{p} \in [0,1]^d$ with $\|\mathbf{p} - \mathbf{q}\|_1 \leq r$, one can produce an estimate $\widehat{\mathbf{p}} \in [0,1]^d$ in $\mathrm{poly}(n,d)$ time such that $\|\widehat{\mathbf{p}} - \mathbf{p}\|_2 \leq \varepsilon$ with success probability at least $1 - \delta$.*

*Proof.* Suppose we get $n$ samples $\mathbf{y}_1, \ldots, \mathbf{y}_n \sim \mathrm{Ber}(\mathbf{p})$. For $i \in [n]$, we can re-express each $\mathbf{y}_i$ as $\mathbf{y}_i = \mathbf{p} + \mathbf{z}_i$ for some $\mathbf{z}_i$ distributed as $\mathrm{Ber}(\mathbf{p}) - \mathbf{p}$. Let us define $\widehat{\mathbf{p}} \in [0,1]^d$ as follows:

$$\widehat{\mathbf{p}} = \operatorname*{argmin}_{\|\mathbf{b} - \mathbf{q}\|_1 \leq r} \frac{1}{n} \sum_{i=1}^{n} \|\mathbf{y}_i - \mathbf{b}\|_2^2 \tag{1}$$

By optimality of $\widehat{\mathbf{p}}$ in Equation (1), we have

$$\frac{1}{n} \sum_{i=1}^{n} \|\mathbf{y}_i - \widehat{\mathbf{p}}\|_2^2 \leq \frac{1}{n} \sum_{i=1}^{n} \|\mathbf{y}_i - \mathbf{p}\|_2^2 \tag{2}$$

By expanding and rearranging Equation (2), one can show:

$$\|\widehat{\mathbf{p}} - \mathbf{p}\|_2^2 \leq \frac{2}{n} \left\langle \sum_{i=1}^{n} \mathbf{z}_i, \widehat{\mathbf{p}} - \mathbf{p} \right\rangle \tag{3}$$

Meanwhile, a standard Chernoff bound shows that $\Pr\left[\|\sum_{i=1}^{n} \mathbf{z}_i\|_\infty \geq \sqrt{2n \log\left(\frac{2d}{\delta}\right)}\right] \leq \delta$. Therefore, using Hölder's inequality and triangle inequality with the above, we see that, with probability at least $1 - \delta$,

$$\|\widehat{\mathbf{p}} - \mathbf{p}\|_2^2 \leq \frac{2}{n} \langle \sum_{i=1}^{n} \mathbf{z}_i, \widehat{\mathbf{p}} - \mathbf{p} \rangle \leq \frac{2}{n} \cdot \left\|\sum_{i=1}^{n} \mathbf{z}_i\right\|_\infty \cdot \|\widehat{\mathbf{p}} - \mathbf{p}\|_1$$

$$\leq \frac{2}{n} \cdot \left\|\sum_{i=1}^{n} \mathbf{z}_i\right\|_\infty \cdot (\|\widehat{\mathbf{p}} - \mathbf{q}\|_1 + \|\mathbf{p} - \mathbf{q}\|_1)$$

$$\leq 4r \cdot \sqrt{\frac{2 \log\left(\frac{2d}{\delta}\right)}{n}}$$

Finally, it is known that LASSO runs in $\mathrm{poly}(n,d)$ time. $\square$

Using Lemma 3.4, we now ready to prove Theorem 3.1.

*Proof of Theorem 3.1.* **Correctness of $\widehat{\mathbf{p}}$ output.** TESTANDOPTIMIZEMEAN (Algorithm 2) has two possible outputs for $\widehat{\mathbf{p}}$:
*Case 1:* $\widehat{\mathbf{p}} = \operatorname{argmin}_{\|\mathbf{b} - \mathbf{q}\|_1 \leq \lambda} \frac{1}{n} \sum_{i=1}^{n} \|\mathbf{y}_i - \mathbf{b}\|_2^2$, which can only happen when Outcome is $\lambda \in \mathbb{R}$ and $\lambda < \varepsilon\sqrt{d}$
*Case 2:* $\widehat{\mathbf{p}} = \frac{1}{n} \sum_{i=1}^{n} \mathbf{y}_i$

Conditioned on APPROXL1 succeeding, with probability at least $1 - \delta$, we will show that $\mathrm{d}_{\mathrm{TV}}(\mathrm{Ber}(\mathbf{p}), \mathrm{Ber}(\widehat{\mathbf{p}})) \leq \varepsilon$ and failure probability at most $\delta$ in each of these cases, which implies the theorem statement.

*Case 1:* Using $r = \lambda$ as the upper bound, Lemma 3.4 tells us that $\|\widehat{\mathbf{p}} - \mathbf{p}\|_2 \leq \varepsilon\sqrt{\tau(1-\tau)}/2$ with failure probability at most $\delta$ when $\widetilde{O}(\lambda^2/\tau^2\varepsilon^4)$ i.i.d. samples are used. Using Proposition 2.4, $\mathrm{d}_{\mathrm{TV}}(\mathrm{Ber}(\mathbf{p}), \mathrm{Ber}(\widehat{\mathbf{p}})) \leq \varepsilon$.

*Case 2:* With $\widetilde{O}(d/\varepsilon^2)$ samples, it is known that the empirical mean $\widehat{\mathbf{p}}$ achieves $\mathrm{d}_{\mathrm{TV}}(\mathrm{Ber}(\mathbf{p}), \mathrm{Ber}(\widehat{\mathbf{p}})) \leq \varepsilon$ with failure probability at most $\delta$.

---

**Algorithm 2** The TESTANDOPTIMIZEMEAN algorithm.

---

1: **Input**: Error rate $\varepsilon > 0$, failure rate $\delta \in (0,1)$, parameter $\eta \in [0, \frac{1}{4}]$, parameter $\tau \in [0, \frac{1}{2}]$, and sample access to $\mathrm{Ber}(\mathbf{p})$
2: **Output**: $\widehat{\mathbf{p}} \in \mathbb{R}^d$
3: Define $k = \min(\lceil d^{4\eta}/\tau^4 \rceil, d)$, $\alpha = \varepsilon d^{(3\eta-1)/2}/\tau$, $\zeta = 4\varepsilon \cdot \sqrt{d}$, and $\delta' = \frac{\delta}{\lceil d/k \rceil \cdot \lceil \log_2 \zeta/\alpha \rceil}$
4: Draw $O(\sqrt{k} \log(1/\delta')/\alpha^2)$ i.i.d. samples from $\mathrm{Ber}(\mathbf{p})$ and store it into a set $\mathcal{S}$
5: Let $\mathtt{Outcome}$ be the output of the APPROXL1 algorithm given $k, \alpha, \zeta$, and $\mathcal{S}$ as inputs
6: **if** $\mathtt{Outcome}$ is $\lambda \in \mathbb{R}$ and $\lambda < \varepsilon\sqrt{d}$ **then**
7:      Draw $n \in \widetilde{O}(\lambda^2/\varepsilon^4)$ i.i.d. samples $\mathbf{y}_1, \ldots, \mathbf{y}_n \in \{0,1\}^d$
8:      **return** $\widehat{\mathbf{p}} = \operatorname{argmin}_{\|\mathbf{b}-\mathbf{q}\|_1 \leq \lambda} \frac{1}{n} \sum_{i=1}^n \|\mathbf{y}_i - \mathbf{b}\|_2^2$
9: **else**
10:      Draw $n \in \widetilde{O}(d/\varepsilon^2)$ i.i.d. samples $\mathbf{y}_1, \ldots, \mathbf{y}_n \in \{0,1\}^d$
11:      **return** $\widehat{\mathbf{p}} = \frac{1}{n} \sum_{i=1}^n \mathbf{y}_i$ {Empirical mean}
12: **end if**

---

**Sample complexity used.** APPROXL1 uses $|\mathbf{S}| = m(k, \alpha, \delta') \in \widetilde{O}(\sqrt{k}/\alpha^2)$ samples to produce $\mathtt{Outcome}$. Then, APPROXL1 further uses $\widetilde{O}(\lambda^2/\tau^2\varepsilon^4)$ samples or $\widetilde{O}(d/\varepsilon^2)$ samples depending on whether $\lambda < \varepsilon\sqrt{d}$. So, TESTANDOPTIMIZEMEAN has a total sample complexity of $\widetilde{O}\left(\frac{\sqrt{k}}{\alpha^2} + \min\left\{\frac{\lambda^2}{\tau^2\varepsilon^4}, \frac{d}{\varepsilon^2}\right\}\right)$. Meanwhile, Lemma 3.3 states that $\|\mathbf{p} - \mathbf{q}\|_1 \leq \lambda \leq 2\sqrt{k} \cdot (\lceil d/k \rceil \cdot \alpha + 2\|\mathbf{p} - \mathbf{q}\|_1)$ whenever $\mathtt{Outcome}$ is $\lambda \in \mathbb{R}$. Since $(a+b)^2 \leq 2a^2 + 2b^2$ for any two real numbers $a, b \in \mathbb{R}$, we see that $\frac{\lambda^2}{\tau^2\varepsilon^4} \in O\left(\frac{k}{\tau^2\varepsilon^4} \cdot \left(\frac{d^2\alpha^2}{k^2} + \|\mathbf{p} - \mathbf{q}\|_1^2\right)\right) \subseteq O\left(\frac{d}{\varepsilon^2} \cdot \frac{1}{\tau^2}\left(\frac{d\alpha^2}{\varepsilon^2 k} + \frac{k \cdot \|\mathbf{p}-\mathbf{q}\|_1^2}{d\varepsilon^2}\right)\right)$. Putting together the above observations, we see that the total sample complexity is

$$\widetilde{O}\left(\frac{\sqrt{k}}{\alpha^2} + \frac{d}{\varepsilon^2} \cdot \min\left\{1, \frac{d\alpha^2}{\varepsilon^2\tau^2 k} + \frac{k \cdot \|\mathbf{p}-\mathbf{q}\|_1^2}{d\tau^2\varepsilon^2}\right\}\right).$$

Recalling that TESTANDOPTIMIZEMEAN sets $k = \min(\lceil d^{4\eta}\tau^{-4}\rceil, d)$ and $\alpha = \varepsilon d^{(3\eta-1)/2}\tau^{-1}$, the above expression simplifies to $\widetilde{O}\left(\frac{d}{\varepsilon^2} \cdot \left(d^{-\eta} + \min\left(1, \frac{\|\mathbf{p}-\widetilde{\mathbf{p}}\|_1^2}{d^{1-4\eta}\tau^6\varepsilon^2}\right)\right)\right)$. □

## 4 Lower Bounds

For proving of our lower bounds, we use the following corollary of Fano's inequality.

**Lemma 4.1** (Lemma 6.1 of [ABDH+20]). *Let $\kappa : \mathbb{R} \to \mathbb{R}$ be a function and let $\mathcal{F}$ be a class of distributions such that, for all $\varepsilon > 0$, there exist distributions $f_1, \ldots, f_M \in \mathcal{F}$ such that*

$$\mathrm{d}_{\mathrm{KL}}(f_i, f_j) \leq \kappa(\varepsilon) \text{ and } \mathrm{d}_{\mathrm{TV}}(f_i, f_j) > 2\varepsilon \ \forall i \neq j \in [M]$$

*Then any method that learns $\mathcal{F}$ to within total variation distance $\varepsilon$ with probability $\geq 2/3$ has sample complexity $\Omega\left(\frac{\log M}{\kappa(\varepsilon)\log(1/\varepsilon)}\right)$.*

**Lemma 4.2** (Learning unbalanced distributions requires linear samples). *Suppose $\varepsilon$ is sufficiently small, and we are given sample access to a product distribution $\mathrm{Ber}(\mathbf{p})$ on $\{0,1\}^d$ with mean vector $\mathbf{p}$ having entries which are $O(1/d)$, along with an advice mean vector $\mathbf{q}$ such that $\|\mathbf{p} - \mathbf{q}\|_1 \leq O(\varepsilon)$. Even in this case, learning $\widehat{\mathbf{p}}$ such that $\mathrm{d}_{\mathrm{TV}}(\mathrm{Ber}(\mathbf{p}), \mathrm{Ber}(\widehat{\mathbf{p}})) \leq \varepsilon$ requires $\widetilde{\Omega}\left(\frac{d}{\varepsilon}\right)$ samples*

*Proof.* Suppose that the advice distribution $\mathrm{Ber}(\mathbf{q})$ has mean vector $\mathbf{q} \triangleq \begin{bmatrix} \frac{\varepsilon}{d} & \cdots & \frac{\varepsilon}{d} \end{bmatrix}$. If $S \subseteq [d]$, define $\mathbf{p}_S \in [0,1]^d$ with $\mathbf{p}_S[i] = \frac{2\varepsilon}{d}$ if $i \in S$ and $= \frac{\varepsilon}{d}$ otherwise. Then, we have $\|\mathbf{p}_S - \mathbf{q}\|_1 = |S|\frac{\varepsilon}{d}$ for all $S \subseteq [d]$. Also, for all $S, T \subseteq [d]$ we have $\mathrm{d}_{\mathrm{TV}}(\mathrm{Ber}(\mathbf{p}_S), \mathrm{Ber}(\mathbf{p}_T)) \geq \left|\mathrm{Pr}_{x\sim\mathrm{Ber}(\mathbf{p}_S)}(x_{S\setminus T} = \mathbf{0}) - \mathrm{Pr}_{x\sim\mathrm{Ber}(\mathbf{p}_T)}(x_{S\setminus T} = \mathbf{0})\right| = \left|\left(1 - \frac{2\varepsilon}{d}\right)^{|S\setminus T|} - \left(1 - \frac{\varepsilon}{d}\right)^{|S\setminus T|}\right| \geq 1 - \left(\frac{|S\setminus T|\varepsilon}{d} + \frac{1}{1+2\frac{|S\setminus T|\varepsilon}{d}}\right)$; this is using the inequalities $(1-x)^r \geq 1 - rx$ for $r \in \{0\} \cup [1, \infty)$,

$x \leq 1$, and $(1-x)^r \leq \frac{1}{1+rx}$ for $r \geq 0$, $x \in (-1/r, 1]$. Using the same argument with the set $T \setminus S$, we get $\mathrm{d_{TV}}(\mathrm{Ber}(\mathbf{p_S}), \mathrm{Ber}(\mathbf{p_T})) \geq 1 - \left( \frac{|T \setminus S| \varepsilon}{d} + \frac{1}{1 + 2\frac{|T \setminus S| \varepsilon}{d}} \right)$. Thus, we have

$\mathrm{d_{TV}}(\mathrm{Ber}(\mathbf{p_S}), \mathrm{Ber}(\mathbf{p_T})) \geq \max_{\substack{H \in \{S \setminus T, T \setminus S\} \\ \xi \triangleq |H| \varepsilon / d}} 1 - \left( \xi + \frac{1}{1 + 2\xi} \right)$. Note that, by calculation, we can show that $1 - \left( \xi + \frac{1}{1+2\xi} \right) \geq \xi/2 - \xi^2$ for $\xi \geq 0$, which is $\Omega(\xi)$ for $\xi \in (0, 1/4)$.

Similarly, we have $\mathrm{d_{KL}}(\mathrm{Ber}(\mathbf{p_S}) \| \mathrm{Ber}(\mathbf{p_T})) = \sum_{i \in [d]} \mathsf{kl}([\mathbf{p}_S]_i, [\mathbf{p}_T]_i) = \sum_{i \in S \setminus T} \mathsf{kl}(\frac{2\varepsilon}{d}, \frac{\varepsilon}{d}) + \sum_{i \in T \setminus S} \mathsf{kl}(\frac{\varepsilon}{d}, \frac{2\varepsilon}{d})$ (where $\mathsf{kl}(p, q) \triangleq \mathrm{d_{KL}}(\mathrm{Ber}(p) \| \mathrm{Ber}(q))$). We can see by simple calculations along with the logarithmic inequality $\ln(1+x) \leq x$ for $x > -1$, that $\mathsf{kl}(\frac{\varepsilon}{d}, \frac{2\varepsilon}{d}) \leq \frac{\varepsilon}{d} \left( 1 - \ln(2) + \frac{\varepsilon}{d - 2\varepsilon} \right) \leq \frac{0.5\varepsilon}{d}$ (for $d \geq 10$ and $\varepsilon \leq 1$), and $\mathsf{kl}(\frac{2\varepsilon}{d}, \frac{\varepsilon}{d}) \leq \frac{\varepsilon}{d} \left( 2\ln(2) - 1 + \frac{\varepsilon}{d - \varepsilon} \right) \leq \frac{0.5}{d}$ (for $d \geq 10$ and $\varepsilon \leq 1$). Thus $\mathrm{d_{KL}}(\mathrm{Ber}(\mathbf{p_S}) \| \mathrm{Ber}(\mathbf{p_T})) \leq \frac{\varepsilon}{2d} (|S \setminus T| + |T \setminus S|) = \frac{|S \oplus T| \varepsilon}{2d}$.

Viewing sets $S \subseteq [d]$ as vectors in $\mathbb{F}_2^d$ and using the Gilbert-Varshamov bound, we can say that, for any constant $c \in (0, 1)$ and sufficiently large $d$, there exists a family of sets $\{S_1, \dots, S_M\} \subseteq 2^{[d]}$ with $M \geq 2^{\Omega(cd)}$ such that $|S_i| = cd$ and $|S_i \oplus S_j| \geq \frac{cd}{4}$ for all $i, j \in [M]$. We use this family to instantiate distributions $f_i \triangleq \mathrm{Ber}(\mathbf{p_{S_i}})$ for each $i \in [M]$.

Suppose we take $S = S_i$, $T = S_j$, with $|S| = |T| = cd$ and $|S \oplus T| \in \left[ \frac{cd}{4}, 2cd \right]$, so that $\mathrm{d_{KL}}(f_i \| f_j) \leq \frac{|S \oplus T| \varepsilon}{2d} \leq c\varepsilon$ for all $i, j \in [M]$. Since $|S_i \oplus S_j| \geq cd/4$, we will have at least one of $H \in \{S \setminus T, T \setminus S\}$ with $|H| \in \left[ \frac{cd}{8}, cd \right]$. Thus, we will have $\frac{|H| \varepsilon}{d} \in \left[ \frac{c\varepsilon}{8}, c\varepsilon \right] = \Theta(\varepsilon)$ (for constant $c > 0$), and $\mathrm{d_{TV}}(f_i, f_j) \geq \Omega(\varepsilon)$ (supposing $c\varepsilon < 1/4$).

By appropriately scaling $\varepsilon$ and applying Lemma 4.1, we can show that we need $\tilde{\Omega}\left( \frac{d}{\varepsilon} \right)$ samples to learn a product distribution $f^* = \mathrm{Ber}(\mathbf{p_{S^*}}) \in \{f_1, \dots, f_M\}$ to within $\varepsilon$ in TV distance, even when given advice $\mathbf{q}$ with $\|\mathbf{p}_{S^*} - \mathbf{q}\|_1 < \varepsilon$, if the distribution mean vector $\mathbf{p}_{S^*}$ are allowed to be unbalanced (specifically, with entries $\leq O(1/d)$). $\qquad \square$

We also prove a sample complexity lower bound for learning product distributions balanced case given advice, which adapts the sample complexity lower bound in ([BCGG25], Lemma 32) for learning multivariate isotropic gaussians $\mathcal{N}(\mu^*, I_d)$ given an advice vector which is close to the true mean vector in $\ell_1$ distance.

**Lemma 4.3.** *Let $\varepsilon > 0$ be sufficiently small. Suppose that we are given sample access to a distribution $\mathrm{Ber}(\mathbf{p})$ where $\mathbf{p}$ is $\frac{1}{4}$-balanced, and also an advice vector $\mathbf{q}$ with $\|\mathbf{p} - \mathbf{q}\|_1 = \lambda \geq 100\varepsilon$. Then, any algorithm that learns $\mathrm{Ber}(\mathbf{p})$ up to distance $\varepsilon$ in total variation with constant failure probability requires $\tilde{\Omega}\left( \min\left\{ \|\mathbf{p} - \mathbf{q}\|_1^2 / \varepsilon^4, d/\varepsilon^2 \right\} \right)$ samples. In particular, when $\|\mathbf{p} - \mathbf{q}\|_1 \geq \varepsilon\sqrt{d}$, we need $\Omega(d/\varepsilon^2)$ samples.*

*Proof.* Suppose we want $\|\mathbf{p} - \mathbf{q}\|_1 = \lambda$ for $\lambda$ sufficiently small. Fix $\mathbf{q} = \begin{bmatrix} \frac{1}{2} & \cdots & \frac{1}{2} \end{bmatrix}$ and suppose $\mathbf{p} = \mathbf{p}_S$ for some $S \subseteq [d]$ with $|S| = k$ such that $\mathbf{p}_S[i] = \frac{1}{2} + \frac{\lambda}{k}$ for $i \in S$ and $= \frac{1}{2}$ otherwise. Then $\|\mathbf{p}_S - \mathbf{q}\|_1 = \lambda$ and $\|\mathbf{p}_S - \mathbf{p}_T\|_2 = \frac{\lambda}{k}\sqrt{|S \oplus T|}$. If $\frac{\lambda}{k} < \frac{1}{4}$, the distributions $\mathbf{p}_S$ are $\tau$-balanced for $\tau = \frac{1}{4}$. In that case, for any $S, T \subseteq [d]$, we can bound $\mathrm{d_{KL}}(\mathrm{Ber}(\mathbf{p_S}) \| \mathrm{Ber}(\mathbf{p_T})) \leq 8 \left( \frac{\lambda}{k} \right)^2 |S \oplus T|$ (by Proposition 2.3), and the total variation distance by $\mathrm{d_{TV}}(\mathrm{Ber}(\mathbf{p}), \mathrm{Ber}(\mathbf{q})) \geq \Omega\left( \min\left\{ 1, \frac{\lambda}{k}\sqrt{|S \oplus T|} \right\} \right)$ (Proposition 2.4).

As in ([BCGG25], Lemma 32), we consider $\{S_1, \dots, S_M\}$ and take $\mathbf{p}_i \triangleq \mathbf{p}_{S_i}$ for a family of $k$-subsets with $M \geq 2^{\Omega(k)}$ and $|S_i \oplus S_j| \geq k/4$ for all $i \neq j$, known to exist via the Gilbert-Varshamov bound. We can do this as long as, e.g. $k \geq 10$. This gives $\mathrm{d_{KL}}(\mathrm{Ber}(\mathbf{p_i}) \| \mathrm{Ber}(\mathbf{p_j})) \leq \frac{16\lambda^2}{k}$ (where $\lambda$ is a function of $\varepsilon$) and $\mathrm{d_{TV}}(\mathrm{Ber}(\mathbf{p_i}), \mathrm{Ber}(\mathbf{p_j})) \geq c\frac{\lambda}{2\sqrt{k}}$ as long as $\frac{\lambda}{2\sqrt{k}} < 1$.

If we choose $k = \lceil \frac{\lambda^2}{\varepsilon^2} \rceil \geq 100$ such that $\lambda = \varepsilon\sqrt{k} < 2\sqrt{k}$, we will get pairwise TV $\geq c\varepsilon/2$ and pairwise KL $\leq \frac{16\varepsilon^2 k}{k} \leq O(\varepsilon^2)$. Finally, scaling $\varepsilon$ before applying Lemma 4.1 gives the result. $\qquad \square$

# 5 Conclusion

This work introduces an efficient algorithm for learning product distributions on the Boolean hypercube when provided with an imperfect advice distribution. The sample complexity of this algorithm is $O(d^{1-\eta}/\varepsilon^2)$ under specific conditions on the advice quality and a "balancedness" assumption on the true distribution. Note that the algorithm's sample complexity becomes sublinear in the dimension $d$ if the advice is sufficiently accurate, and it remains robust even with poor advice. Key to this is a novel tolerant mean tester and techniques for approximating the $\ell_1$-distance between the true and advice distributions. Future research could extend this learning-with-advice framework to other complex models like Bayesian networks and Ising models, aiming to understand how structural properties of these models interact with advice quality. It would also be interesting to investigate if advice can improve the sample complexity of learning an unstructured distribution over a discrete domain $[n]$ compared to the classical upper bound of $O(\frac{n}{\varepsilon^2})$ samples.

## Acknowledgments and Disclosure of Funding

PGJ's research is supported by the National Research Foundation, Prime Minister's Office, Singapore under its Campus for Research Excellence and Technological Enterprise (CREATE) programme. TG's research is supported by a start up grant at Nanyang Technological University.

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
