# OpenReview forum: "Product Distribution Learning with Imperfect Advice"
_NeurIPS.cc/2025/Conference — NeurIPS 2025 spotlight_

### Official Review · Reviewer_84iq · 2025-06-28

**Clarity:** 3
**Significance:** 3
**Originality:** 3
**Rating:** 5
**Confidence:** 3

**Summary:**

The paper falls in the line of research of algorithms with predictions where algorithms are equipped with advice or predictions. Given recent developments in machine learning it is often too pessimistic to run a worst-case algorithm and not leverage predictable structures of the input. The line of research seeks to bridge algorithms and ML by obtaining provably improved algorithms with accurate predictions while also maintaining worst case guarantees.

The paper studies the problem of learning a product distribution $\bf{p}$ (parametrized by Bernoulli parameters $(p_1,\dots,p_d)$) over the hypercube $\{0,1\}^d$. It is well-known that $\Theta(d/\varepsilon^2)$ is necessary and sufficient in the classic setting of the problem where we want to return a distribution of total variation distance at most $\varepsilon$ to $\textbf{p}$. The paper shows that if each $p_i\in [\tau,1-\tau]$ (they call this assumption \emph{balancedness}), say for some $\tau=\Omega(1)$, if we are given access to an \emph{advice} distribution $\bf{q}$ such that $\|\textbf{p}-\textbf{q}\|_1$ is small, one can reduce the sample complexity. Specifically, one can obtain sublinear sample complexity $\tilde O (d^{1-\eta}/\varepsilon^2)$ as long as $\|\textbf{p}-\textbf{q}\|_1\leq \varepsilon d^{0.5-\Omega(\eta)}$. The paper additionally shows lower bounds demonstrating that without the balancedness assumption, even with very accurate advice, say $\|\textbf{p}-\textbf{q}\|_1=O(1)$, we must use $\Omega(d/\varepsilon)$ samples. Additionally, they show that when $\|\textbf{p}-\textbf{q}\|_1\geq \varepsilon\sqrt{d}$, we again require $\Omega(\sqrt{d})$ samples.

I found the problem well motivated and the techniques quite interesting. The main challenge is to estimate $\|\textbf{p}-\textbf{q}\|_1$ using a small number of samples from $\textbf{p}$. If this quantity is large, say $\geq \lambda$, the algorithm just resorts to its classic counterpart which returns the empirical mean of a bunch of samples. Otherwise, they can return the solution to a constrained (by $\|\hat \textbf{p}-\textbf{q}\|\leq \lambda$) least squares problem. With this constraint, the sample complexity is correspondingly smaller.

**Questions:**

l89: You say that we can simply return $\textbf{q}$ but this sound different from what your algorithm does, namely either running the least squares algorithm or the classic algorithm. Can you clarify?

l155: $\|\textbf{p}\|\to \|\textbf{p}\|_1$

l159: $m(k,\alpha, \delta')$ hasn't been defined.

Algorithm 1, l6: Do  you want to define $\mathcal{S}_j$ as a multiset?

Algorithm 1, l11: It's not really clear what it means to run the algorithms with these parameters. For example, you should mention that $l_i$ is the accuracy paramter.

l174: $n_{k,\varepsilon}$ and $r_\delta$ haven't been defined.

l181: I think you want to claim that $\|\textbf{p}\|\to \|\textbf{p}\|_2$ is large

l187: What does it mean to perform a constrained LASSO?

And finally: The dependence on $\varepsilon$ seems to be $1/\varepsilon$ in your lower bound in Lemma 4.2, but the upper bound is $O(d/\varepsilon^2)$. Do you have any idea what the right dependence is?

**Ethical Concerns:**

["NO or VERY MINOR ethics concerns only"]

**Final Justification:**

Please see the discussion with the authors

**Quality:**

3

**Strengths And Weaknesses:**

(1) The paper is well and clearly written and I had little difficulty following the proofs.
(2) Distribution learning is an important problem and I found the  model well-motivated and interesting.
(3) The algorithmic/theoretical content of the paper is quite nice. It seems to me that the main algorithmic contribution is the algorithm \texttt{ApproxL1} which fails if $\|\textbf{p}-\textbf{q}\|_2$ is large (which implies that $\|\textbf{p}-\textbf{q}\|_1$ is large) and otherwise outputs an estimate $\lambda$ of $\|\textbf{p}-\textbf{q}\|_1$.

\textbf{Weaknesses} (1) The author's state that their paper and analysis parallels that of [BCJG24] which address the problem of learning Gaussian distributions with imperfect advice. They state that one of their different contributions is their new $\ell_2$-approximation algorithm. If by this, they refer to the one of Lemma 3.2, this is maybe a little disappointing as the ideas are very similar to [DK16]. I am thus curious whether the ideas behind Lemma 3.3 are new, or if similar ideas appeared in [BCJG24].

---

> ### Author Rebuttal · Authors · 2025-07-31
>
> Thank you for the time and effort to review our work, and for your feedback.
>
> **Lemma 3.2:**
> While we agree that the test statistic is taken from [DK16], the context is quite different, as the authors there were studying 1-dimensional distributions,
> whereas here, we apply the test to high-dimensional product distributions. Also, there is no additional flattening step, as is necessary to get the optimal
> sample complexity in the 1-dimensional case.
>
>
> **Line 89:**
> You are right that we did not describe Algorithm 2 as such. We can explicitly write an additional case in our algorithm (where the advice is good enough to return directly without a Lasso optimization step). But it does not qualitatively change our theoretical guarantees, so we opted to absorb this special case into the least-squares Lasso optimization case.
>
> **Line 155:**
> Thank you for pointing out the typo. In Lines 155 and 156, it should be $\|p\|_2$ and $\|p-q\|_2$. We will add fix these in our revision.
>
> **Line 159 and 174:**
> We take $n_{k, \varepsilon} = \lceil\frac{16\sqrt{k}}{3\varepsilon^2}\rceil$, $r_{\delta} = 1 + \lceil \log(12/\delta) \rceil$, and define $m(k, \alpha, \delta) \triangleq n_{k, \varepsilon} \cdot r_{\delta}$ for any choice of $\varepsilon, \delta$. We will add these definitions properly in our final version. As in lines 174-177, we set $\delta$ to $\delta^\prime = \frac{\delta}{w \cdot \lceil \log_2 (\zeta/\alpha)\rceil}$ in each invocation and then take a union bound over all invocations. Hence we use $m(k, \alpha, \delta^\prime)$ samples.
>
> **Algorithm 1, Line 6:**
> Yes, both $\mathcal{S}$ and $\mathcal{S}_j$ are indeed multisets.
>
> **Algorithm 1, Line 11:**
> We invoke ApproxL1 (Lemma 3.2) with accuracy $\varepsilon = \ell_i$, $\delta = \delta'$ (as defined in line 176), $S_j$ as samples from $\mathrm{Ber}(\mathbf{p})$ projected to coordinates $B_j$, and $\mathbf{q}$ projected to coordinates $B_j$ as the advice. We will write this mapping more clearly in our revision.
>
> **Line 181:**
> Sorry, could you please clarify what you mean by $\|p\| \to \|p\|_2$ here? There is no mention of $p$ on Line 181.
>
> **Line 187:**
> By constrained LASSO, we meant the optimization problem:
>
> $\arg\min_{\|\mathbf{b}-\mathbf{q}\|_1 \leq \lambda} \frac{1}{n} \sum_{i=1}^{n} \|\mathbf{b} - \mathbf{q}_i\|_2^2$
>
>
> which is what we put in Line 8 of Algorithm 2. This is the standard version of LASSO with an $\ell_1$ constraint. We will change the phrasing to ``LASSO in its constraint form''.
>
> **Dependence on $\varepsilon$ in the lower bound and upper bound:**
> Lemma 4.2 is meant to show the linear dependence on $d$ in the unbalanced setting. The $1/\varepsilon^2$ dependence comes from the balanced setting. Lemma 4.3 for the balanced setting is currently written for constant $\varepsilon$. In the final version, we will put in a more refined argument that shows the correct dependence on $\varepsilon$ as well; this is similar to the lower bound for learning gaussians.

---

> > ### Comment · Reviewer_84iq · 2025-08-03
> >
> > Thanks for the feedback and for the nice reading experience of the paper. I retain my score.
> >
> > For l181: I'm not sure what I was referring to. I probably meant a different line, but it seems to be a minor typo in any case.

---

### Official Review · Reviewer_Ch6Q · 2025-07-01

**Clarity:** 3
**Significance:** 3
**Originality:** 3
**Rating:** 4
**Confidence:** 1

**Summary:**

The authors considered the problem setting of learning product distribution under imperfect advice,
With some assumption on the mean for each coordinate that is balanced (not too close to 0 or 1).
The problem setting is as follows.
Given an unknown product distribution of {0,1}^d (with p be the mean vector of it),
and advice vector q.
The algorithm needs to learn a $\hat{p}$ as an estimation of p, such that the product distribution specified by $\hat{p}$
It is close to the true input product distribution in total variance distance.
(With the balanced assumption here, this is equivalent to small l_2 norm between $\hat{p}$ and $p$.)
If the given advice is good ($\|q-p\|_1$ is small), the algorithm needs to use fewer samples than the trivial $d/\epsilon^2$ sample complexity.
And if the given advice is bad, the algorithm needs to perform as well as the trivial $d/\epsilon^2$ sample complexity.
This need to be done without the previous knowledge of $\|q-p\|_1$.

For their result, the authors give a polynomial time algorithm with $d^{1-\eta}/\epsilon^2$ sample complexity algorithm when $\|p-q\|_1<\epsilon d^{0.5-\Omega(\eta)}$.
The high-level idea they used is the following:
The first step is to get an estimation of $\|q-p\|_1$.
Directly approximating it would result in linear sample complexity and defeat the purpose.
So the author partitions it into d/k blocks, each with k coordinates.
Then, estimate the L2 norm difference on each block and add them up.
This gives an estimation of the l1 norm by a multiplicative error factor of at most $\sqrt{k}$.
After that, the problem can then be reformulated as the convex optimization below line 84.

I feel like the result is far from optimal, and a lot more should be done in this setting.
In particular, there isn't much justification from the lower bound side, or some argument such as
this result is the best achievable under some particular type of methods.
Furthermore, it seems both the assumption and the guarantee are very artificial.
(mostly to incorporate the methodology using L2 norm from my understanding)
The author didn't explain about this L2 norm approach, like whether they are typical in this setting, or if this is the best we can do.
This is based on my limited knowledge of this topic, and I would encourage the authors to point out any mistakes I may have made.

**Questions:**

Can the authors explain more about why the assumptions are need here?
Especially if there is anything known about the sample complexity for unbalanced product distribution
when the advice has o(1) l1 error.

**Ethical Concerns:**

["NO or VERY MINOR ethics concerns only"]

**Limitations:**

Yes.

**Quality:**

3

**Strengths And Weaknesses:**

One weakness I can think of is that the balanced assumption isn't very well justified.
While the author did mention that there no sublinear sample complexity algorithm even when advice is O(1) close
to the truth in l1 distance, it is unclear how sample complexity changes with o(1) l1 distance.
And it is plausible that less samples are needed given the previous $\sqrt{n}/\epsilon^2$ indentity testing result.

---

> ### Author Rebuttal · Authors · 2025-07-31
>
> Thank you for time and effort to review our work, and for your feedback.
>
>
> **Use of $\ell_2$ norm:**
> We would like to counter the suggestion that the $\ell_2$ norm is artificial in this setting. It is known that the $\ell_2$-distance between the mean
> vectors is a lower bound on the TV distance between product distributions (Kontorovich '25); so, to learn product distributions with small TV error, it is
>  *necessary* to learn the mean vector with small $\ell_2$ error.
>
> **Justification from the lower bound side:**
> To be clear, in Lemma 4.2, the condition $\|p-q\|_1<\epsilon$ implies that the total variation distance between the product distributions with mean vectors $p$ and $q$
> is $<\varepsilon$. This is also from (Kontorovich '25). So, Lemma 4.2 makes sense only if $\|p-q\|_1>\varepsilon$, since otherwise the sample complexity would become 0
> as an algorithm could simply return $q$.
>
> Note that our lower bounds are in the easier setting where the $\ell_1$ distance from the advice and true mean vectors is promised to satisfy an upper bound.
> Our algorithms are in the harder setting where there is no promise, so our lower bounds automatically translate. Your question about learning when
> the advice quality is $\ll \varepsilon$ is definitely valid when the advice error is not known to the algorithm, and we have not explored lower bounds in this setting here.
>
> We will make sure to add a fuller discussion of these issues in the final version.
>
> **References**
> 1. Aryeh Kontorovich. ``On the tensorization of the variational distance.'' Electronic Communications in Probability, 30:1–10, 2025.

---

> > ### Author Response · Authors · 2025-08-08
> > **Follow-up**
> >
> > Hope we addressed your concerns about the lower bound and use of the $\ell_2$-norm. Otherwise, please let us know if we can clarify further. Thank you again for your helpful comments.

---

### Official Review · Reviewer_kA8u · 2025-07-03

**Clarity:** 3
**Significance:** 3
**Originality:** 3
**Rating:** 5
**Confidence:** 3

**Summary:**

This paper studies distribution learning with advice, within the line of research on learning-augmented algorithms or algorithms with predictions. In the standard version of the problem, the algorithm is given samples from a product distribution $P$ over $\{0,1\}^d$ and the goal is to output an estimate of $P$ which is $\epsilon$ close in TV distance. In this work, the algorithm is also given access to the parameters of another distribution $Q$, also a product distribution over $\{0,1\}^d$. The goal is to design an algorithm which is more sample efficient if $Q$ is close to $P$ and which is not much more sample inefficient ($O(d/\epsilon^2)$ samples is the benchmark in the standard setting) for worst-case $Q$.

The main result of this paper is an algortihm which improves over $O(d/\epsilon^2)$ samples as long as $\|Q-P\|_1 \ll \epsilon \sqrt{d}$. This result is restricted to the case where every coordinate of $P$ has probability bounded away from $0$ and $1$. The authors show that both this condition and the dependence on the $\ell_1$ distance between $P$ and $Q$ are required: there exist distributions which do not satisfy these conditions for which sublinear in $d$ samples are insufficient for learning.

If the $\ell_1$ distance between $P$ and $Q$ was known, then the authors show that a reduction to hypothesis selection by covering the $\ell_1$ ball around $Q$ with $\ell_2$ balls of radius $\epsilon$. To estimate the distance, the authors develop a tolerant tester for product distributions based on tolerant testers for discrete distributions over $[d]$.

**Questions:**

Is there hope to improve the quantitative dependence on the $\ell_1$ distance?

**Ethical Concerns:**

["NO or VERY MINOR ethics concerns only"]

**Final Justification:**

I maintain my positive score for the reasons given in the review.

**Limitations:**

Yes.

**Quality:**

3

**Strengths And Weaknesses:**

This paper addresses the natural and fundamental problem of learning product distributions over the Boolean hypercube. The advice in the model is natural and the main result shows a nice tradeoff between the accuracy of the prediction and sample complexity of the algorithm.

The dependence on the $\ell_1$ norm and the restriction to balanced distributions are justified by hard examples. Despite this, the balanced assumption in particular seems somewhat strong. It would be very interesting if some relaxations of this assumption could be made which avoid the hard example and still allow for sublinear learning when the prediction is good.

The paper is purely theoretically, and I would be very interested to see (proof-of-concept) experiments which could perhaps investigate whether $\ell_1$ distance between $p$ and $q$ is strongly correlated empirically with the accuracy of perhaps a simplified version of your tester.

## Minor Comments

Lemma 4.2 is missing a period at the end of the lemma.

---

> ### Author Rebuttal · Authors · 2025-07-31
>
> Thank you for time and effort to review our work, and for your feedback.
>
> **Balancedness assumption:**
> In Lemma 4.2, we have shown that learning unbalanced distributions (with $O(1/d)$ entries in the mean vector) requires a number of samples linear in the dimension $d$, even when given ``very good'' advice (with $O(\varepsilon)$ $\ell_1$ distance). Such an assumption is also common in the product distribution learning literature (e.g. Freund and Mansour 1999, Theorem 4.14). Such an assumption can be avoided, for instance by using the reduction (taking mixture size $k = 1$) from learning the parameters in $\ell_\infty$ distance to learning a (mixture of one) product distribution in KL divergence, which can be found in (Feldman et al. , Theorems 16 and 17). But then the number of samples required is linear (in dimension $d$). This defeats the purpose of using sublinear samples (in $d$) when given sufficiently good advice.
>
>
> **Proof of concept experiments:**
> This paper is intended as a primarily theoretical work. However, the algorithm is based on an elementary test statistic (described in Lemma 3.2), the exponential search with coordinate-bucketing in ApproxL1, and the standard LASSO algorithm (implemented in packages like scikit-learn, etc). We can include some basic experiments in the final version of the paper.
>
>
> **Improving quantitative dependence on the $\ell_1$ distance:**
> Under balancedness, our lower bound in Lemma 4.3 shows that the dependency on $d$ is tight for constant $\varepsilon$.
>
> **References**
>
> 1. Y. Freund and Y. Mansour. ``Estimating a mixture of two product distributions''. COLT 1999.
>
> 2. J. Feldman, R. O'Donnell and R. Servedio, ``Learning mixtures of product distributions over discrete domains''. SIAM Journal on Computing 2008.

---

> > ### Comment · Reviewer_kA8u · 2025-08-04
> > **Response to Author Rebuttal**
> >
> > Thanks for addressing my questions.

---

### Official Review · Reviewer_kAA6 · 2025-07-03

**Clarity:** 3
**Significance:** 3
**Originality:** 3
**Rating:** 4
**Confidence:** 3

**Summary:**

I really like the model of "learning with imperfect advice", which I think is both conceptually interesting and practically well motivated.  In my own words, and at a high level,  this model is the following: A learner interacts with some untrusted third party which gives them "advice". If the advice is "good" then the query complexity of the learner should improve over the baseline learning algorithm. However, the learner should never require more queries than the baseline algorithm, even if the advice is "bad" -- i.e. the learning algorithm should be robust against "bad" advice. The learner has no promise on whether the advice is good or bad.
Given this setup, there is a clear high-level algorithm for this problem, which is roughly the following:
Test if the advice is good or bad.
If the advice is good, accept the advice - you're done.
If the advice was bad, revert to learning, possibly using extra information gained during testing.
From the above its clear that this model mixes learning and testing in an interesting way,  and that learning with sublinear query complexity should be possible when the advice is good and a sublinear test for this is possible. As before, I think this really is an interesting model, and that there are many many natural and interesting open questions.
With this in mind, the submitted manuscript studies the problem of learning product distributions in this model, where the advice is a full description of a candidate product distribution. This is a natural analogue of previous work in this area on learning multivariate Gaussians with imperfect advice (BCGJ24), and indeed the high-level structure of the algorithm given in the manuscript is analogous to the one given in BCGJ24. However, to instantiate this algorithm the authors of the submitted manuscript need to develop a novel tolerant mean tester and l1 approximation algorithm. Interestingly, it also turns out that unlike the multivariate gaussian case, one provably needs a "balancedness" assumption in this setting (even though this assumption is not necessary for identity testing of product distributions!). With this in mind, I think the manuscript has the following strengths and weaknesses:

**Questions:**

None.

**Ethical Concerns:**

["NO or VERY MINOR ethics concerns only"]

**Limitations:**

None.

**Paper Formatting Concerns:**

None.

**Quality:**

3

**Strengths And Weaknesses:**

Strengths:
I think that the model considered is very interesting and well motivated, and the paper provides some foundational results in this model, which has only very recently been proposed and studied for the first time. I think its great to draw more attention to this model!
The paper exposes an interesting and unexpected  discrepancy between testing product distributions and learning product distributions with imperfect advice.

Weaknesses:
In many ways (both upper and lower bounds) the submitted manuscript parallels the earlier work of BCGJ24 on learning multivariate gaussians in this model. While the authors do require some novel elements -- namely a new tolerant mean tester and l1 approximation algorithm -- the high level ideas mostly already present in earlier work. To be fair though, because there is an "obvious" high-level algorithm, most work in this area will probably share the same key ideas and methods.

---

> ### Author Rebuttal · Authors · 2025-07-31
>
> Thank you for your time and effort to review our work and for your feedback. While our work indeed closely follows the framework of [BCJG24] as you said, there
> are both conceptual as well as technical differences. Conceptually, [BCJG24] uses tools from linear algebra and facts about probability densities, while the
> work here is for discrete distributions on the Boolean hypercube. Technically, the novelty here is in (i) the tolerant mean-vector L2 tester and using it to approximate mean-vector L1 distance for binary product distributions, (ii) identifying the necessity of the balancedness assumption (already explored in the product distribution learning literature) in learning using sublinear (in dimension $d$) samples, even when given good advice (Lemma 4.2).

---

### Decision · Program_Chairs · 2025-09-17

**Decision:**

Accept (spotlight)

**Comment:**

This paper studies the classic and fundamental problem of learning a discrete product distribution $p$ over $\{0,1\}^d$ from i.i.d. samples, with the twist that the algorithm is now given some additional (imperfect) advice $q$ on what the true distribution is. While the classic problem is well-understood, requiring $\Theta(d/\epsilon^2)$ samples to learn to total variation distance $\epsilon$, this paper shows how to beat the bound polynomially under a balancedness assumption---which the authors show are necessary---and improve the sample complexity to $O(d^{1-\eta}/\epsilon^2)$ where $\eta$ is a parameter derived from how close $p$ and $q$ are.

Reviewers and I all agree that this paper has an interesting model and an interesting result, and is a solid contribution to NeurIPS.